# Normal-Fat vs. High-Fat Diets and Olive Oil vs. CLA-Rich Dairy Fat: A Comparative Study of Their Effects on Atherosclerosis in Male Golden Syrian Hamsters

**DOI:** 10.3390/metabo13070827

**Published:** 2023-07-06

**Authors:** Alaitz Berriozabalgoitia, Juan Carlos Ruiz de Gordoa, Gustavo Amores, Gorka Santamarina-Garcia, Igor Hernández, Mailo Virto

**Affiliations:** 1Lactiker Research Group, Department of Biochemistry and Molecular Biology, University of the Basque Country UPV/EHU, Paseo de la Universidad 7, 01006 Vitoria-Gasteiz, Spain; alaitzberrio@gmail.com (A.B.); jcruizdegordoa@hotmail.com (J.C.R.d.G.); gustavo.amores@ehu.eus (G.A.); gorka.santamarina@ehu.eus (G.S.-G.); igor.hernandezo@ehu.eus (I.H.); 2Bioaraba, Prevención, Promoción y Cuidados en Salud, 01009 Vitoria-Gasteiz, Spain

**Keywords:** golden Syrian hamsters, milk fat, olive oil, lipid metabolism, cholesterol, atherogenic lesions, vaccenic acid, rumenic acid, LDL-oxidation

## Abstract

The relationship between milk fat intake (because of its high saturated fatty acid content) and the risk of suffering from cardiovascular diseases remains controversial. Thus, Golden Syrian hamsters were fed two types of fat—sheep milk fat that was rich in rumenic (*cis*9,*trans*11-18:2) and vaccenic (*trans*11-18:1) acids and olive oil—and two doses (a high- or normal-fat diet) for 14 weeks, and markers of lipid metabolism and atherosclerosis evolution were analyzed. The results revealed that the type and percentage of fat affected most plasma biochemical parameters related to lipid metabolism, while only the expression of five (CD36, SR-B1, ACAT, LDLR, and HMG-CoAR) of the studied lipid-metabolism-related genes was affected by these factors. According to aortic histology, when ingested in excess, both fats caused a similar increase in the thickness of fatty streaks, but the high-milk-fat-based diet caused a more atherogenic plasma profile. The compositions of the fats that were used, the results that were obtained, and the scientific literature indicated that the rumenic acid present in milk fat would regulate the expression of genes involved in ROS generation and, thus, protect against LDL oxidation, causing an effect similar to that of olive oil.

## 1. Introduction

Although cardiovascular diseases (CVDs) are multifactorial, the most widely accepted parameters related to the risk of their development are the total plasma cholesterol (TC) and lipoprotein concentrations, especially low-density lipoprotein cholesterol (LDL-C) concentrations. Thus, it has been long established that the amount and type of fat in a diet profoundly influence plasma lipid levels and lipoprotein metabolism [1].

Dairy products comprise a group of heterogeneous food products that consists primarily of milk, cheese, and yogurts from ruminants—mostly cows, sheep, and goats. They provide a large number of essential nutrients that can be beneficial for most people. However, their positive effects on health have been questioned given their high saturated fatty acids (SFA) content. Thus, the scientific evidence is still controversial, as some authors have concluded [2] that milk and dairy products may protect against chronic diseases, while other authors have stated that their association with a reduced risk of developing CVDs is still controversial [3]. One reason may be the variability in milk fat (MF) fatty acid (FA) composition, which depends on factors such as the animal species or feeding regime. The feeding regime has a special influence on the concentration of minor FAs, such as vaccenic acid (VA, *trans*11-18:1) and conjugated linoleic acids (CLAs), especially rumenic acid (RA, *cis*9,*trans*11-18:2). The latter is associated with increased protection against CVDs. VA and CLAs are intermediate products of the microbial biohydrogenation of linoleic acid into stearic acid in the rumen of animals [4]. The diet of a dairy animal is the main factor affecting the CLA content of its milk. Thus, for example, it has been proven that grazing improves the cardiovascular (CV) features of MF by increasing its VA and RA concentrations, as well as its polyunsaturated FA (PUFA) content [5].

On the other hand, olive oil (OO) is high in oleic acid (OA), which is a monounsaturated FA (MUFA), and it has been proven to reduce the risk of CV events [6]. In the same way, OO possesses minor components, such as vitamin E and phenols, which protect LDL against oxidation, thus decreasing the risk of developing atherosclerosis [7].

Golden Syrian hamsters (*Mesocricetus auratus*) began to be used to study diet-induced atherosclerosis and CVDs in the 1980s. It has been proven that they are a suitable model for research on diseases of this type, as they tend to suffer from hypercholesterolemia and early atherosclerotic lesions with an atherogenic diet [8]. Similar to humans, hamsters have a low rate of endogenous cholesterol synthesis, cholesteryl ester transfer protein (CEPT) activity, hepatic apoB-100 secretion, and uptake of the majority of LDL-C via the LDL receptor (LDLR) pathway [9]. The pathological processes that lead to the development of atherosclerosis involve the accumulation of lipids in the arterial wall from plasma lipoproteins, and diet is one of the factors with the greatest influence on this process [10]. However, over recent decades, considerable variability has been reported regarding the responsiveness of hamsters to diet-induced atherosclerosis. This variability may be attributable to differences in hamster strain, background diet, and dietary fat type [11].

For all of these reasons, a study was proposed in which the same strain of hamsters was subjected to a diet with normal fat content (7%, *w*/*w*), and this group was compared with hamsters that were subjected to a high-fat diet (21%, *w*/*w*), which has been described as atherogenic for hamsters [12]. At the same time, the effects of two types of fat (MF and OO), which have very different FA contents and have been associated with opposite effects on the development of atherosclerosis, were compared. The main objective of this study was to analyze the metabolic response of these animals to these diets and to study the ways in which lipid metabolism and atherosclerotic outcomes with RA-rich MF cause a differential response in comparison with outcomes with OO. To the best of our knowledge, no comparable studies have been carried out on hamsters so far.

## 2. Materials and Methods

### 2.1. Hamsters, Diets, and Experimental Design

The experiments with hamsters were carried out in accordance with the institution’s guidelines for the care and use of laboratory animals (approval document reference: CEBA/209/2011/VIRTO LECUONA). Four-week-old male Golden Syrian hamsters (*Mesocricetus auratus*; RjHan:AURA; Janvier Europe, Le Genest-Saint-Isle, France) (n = 32) were housed at controlled room temperature and humidity and under a 12:12 h artificial light/dark cycle (light on at 21:00) with free access to water and food. After an adaptation period of one week, the hamsters were randomly distributed into four experimental groups of eight animals each, with two animals in each cage. Each group followed an experimental diet for 14 weeks. Basal mix diets were prepared by Harlan (Teklad Custom Research Diet, Harlan Laboratories, Madison, WI, USA) and were supplemented with OO or MF to obtain final concentrations of 7% (normal-fat diets: OO7 and MF7) and 21% (high-fat diets: OO21 and MF21) fat by weight. Cholesterol was added to the OO7 and OO21 diets in the concentration needed to obtain the same concentration that is naturally present in MF. Commercial OO was purchased from a local market (Virgin Olive Oil, La Española, Sevilla, Spain). Milk cream was obtained through centrifugation (2000× *g* at 4 °C for 30 min) of raw sheep milk purchased from a local sheepherder. Sheep milk was used because of its higher fat, VA, and RA contents compared with cow milk [5]. The general compositions of diets and the FA compositions of the fats are shown in Table 1 and Appendix A, respectively.

Hamsters were feed-deprived for 12 h and anesthetized with isoflurane (100% *w*/*w*, Esteve, Barcelona, Spain) before sacrifice. Blood samples were collected through cardiac puncture with capillary tubes, with EDTA as an anticoagulant. Plasma was obtained through centrifugation at 800× *g* for 30 min and kept frozen at −80 °C until the analysis.

### 2.2. Plasma Fatty Acid Analysis

Fat extraction from the plasma and total FA methylation were accomplished in one step according to a modified method from Bondia-Pons [13,14]. FA methyl esters (FAMEs) were prepared from 200 µL of plasma by sequentially adding 2.5 mL of 0.5 M sodium methoxide in methanol, followed by 2.5 mL of 14% boron trifluoride in methanol. The resulting FAMEs were extracted by adding 1.0 mL n-hexane and collected in a vial with anhydrous sodium sulfate. FAME separation was achieved through gas chromatography on a CpSil88 capillary column (100 m × 0.25 mm i.d. × 0.20 μm film thickness; Varian Inc., Palo Alto, CA, USA). The chromatograph (Agilent Technologies, Madrid, Spain) was equipped with an FID detector. The FAMEs were identified by comparing the retention times of the obtained peaks with those of authentic standards. The internal standard method was used to quantify the amount of each compound, with undecanoic (11:0), tridecanoic (13:0), and nonadecanoic (19:0) acids as internal standards. The absolute concentration of FA was expressed in µmol/L, and from this, the molar percentages were calculated.

### 2.3. Plasma Cholesterol, Triglyceride, Glucose, and Antioxidant Analyses

The plasma biochemical parameters were analyzed via colorimetric enzymatic assays with commercial kits; fasting glucose, triglycerides (TGs), and total cholesterol (TC) were assessed with Biosystems kits (Biosystems, Barcelona, Spain); and a Wako kit was used for free cholesterol (FC) (Wako Chemicals, Richmond, VA, USA). Cholesteryl esters (CEs) were calculated by subtracting the free cholesterol from the total cholesterol.

For plasma antioxidant extraction, one volume of plasma was first deproteinized with one volume of ethanol, and then, tocopherols and retinoids were extracted twice with two volumes of hexane, as previously described [15]. Tocopherols and retinoids were simultaneously separated using normal-phase HPLC with a Zorbax RX-SIL column (25 cm × 4.6 mm i.d.; 5 μm particle size; Agilent Technologies, Madrid, Spain) installed on a 2695 Alliance separation module (Waters, Barcelona, Spain) coupled with a fluorescence detector (model 474, Waters), as described previously [16].

### 2.4. Liver and Feces Lipid Extraction and Hepatic Lipid Analysis

Lipids from the liver and feces were extracted via the modified Folch method, as described previously [17]. A portion of 100 mg of liver was homogenized with 50 mg of sodium sulfate. The feces were ground in a mortar. In total, 4 mL of methanol was added, and the mixture was homogenized a second time, followed by the addition of 8 mL of chloroform. Then, 3 mL of a solution containing 1.25% KCl and 0.05% H_2_SO_4_ was added and centrifuged at 400× *g*, at room temperature, for 10 min. The bottom layer was reserved, and the supernatant was re-extracted with 3 mL of chloroform/methanol (2:1) and centrifuged as previously described. The bottom layer was transferred and pooled with the previous one. The pooled solution was dried at 37 °C under N_2_.

The liver TC, FE, and TG concentrations were determined enzymatically by using the same kits as those described for the plasma measurements. CE was calculated by subtracting the FC from the TC.

### 2.5. RNA Extraction and Gene Expression Analysis

The total RNA was extracted from a portion of the liver (approximately 30 mg) with a NucleoSpin^®^ RNA plus kit (Macherey-Nagel, Düren, Germany) according to the manufacturer’s instructions. Reverse transcription was carried out with a PrimerScript^TM^ RT reagent kit (Takara Bio Europe SAS, Saint Germain en Laye, France) according to the manufacturer’s protocol. Primers for the amplification of the β-actin, peroxisome proliferator-activated receptor α (PPAR-α), HDL scavenger receptor BI (SR-B1), acyl-CoA: cholesterol acyltransferase (ACAT), microsomal triglyceride transfer protein (MTP), hydroximethylglutaryl-CoA reductase (HMG-CoAR), cluster of differentiation 36 (CD36), liver X receptor (LXR), ApoB100, LDLR, and ApoA1 genes of *Mesocricetus auratus* are described in Appendix A. They were found in the literature or designed with the PerlPrimer vt1.21 software (Parkville, Australia) while taking gene sequences published in the NCBI database (National Center of Biotechnology Information, U.S. National Library of Medicine, Bethesda, MD, USA) into account. The mRNA levels were quantified with an ABI PRIMS^®^ 7000 termocycler (Thermo Scientific, Foster City, CA, USA), and the results were analyzed with the 7000 System ISDS software (v1.2.3.) (Thermo Scientific, Foster City, CA, USA). β-actin was used as the reference gene, and gene expressions were calculated via relative quantification by using the 2-ΔΔCt method [18].

### 2.6. Analysis of Atherosclerotic Lesions in the Hamster Aorta

After exsanguination, the heart and the upper section of the aorta were removed, and the aorta arch was cut, perfused, and stored in a 4% formalin (Panreac, Barcelone, Spain) and 10% sucrose (Merck, Darmstadt, Germany) solution. For analysis, histological sections were frozen in an embedding medium (Tissue-Tek O. C. T., Sakura Finetech USA, Inc., Torrance, CA, USA) and cut by means of a Leica CM30 cryostat (Leica Geosystems, Madrid, Spain). Sections of approximately 10 µm were prepared and mounted on microscope slides. The slides were allowed to dry overnight at room temperature. Then, they were stained with Oil Red O (Sigma-Aldrich, Madrid, Spain) to identify neutral lipid accumulations and counterstained with hematoxylin, as described by Cholewiak (1986) [19].

Images were captured and analyzed with an Olympus BX50 fluorescence photomicroscope (Tokyo, Japan).

### 2.7. Statistical Analysis

Data were expressed as mean ± SD and were analyzed using the one-way Kruskal–Wallis nonparametric ANOVA test (followed by Dunn’s multiple comparisons test). Statistical significance was declared at *p* ≤ 0.05. The analyses were performed with the IBM-SPSS statistical software for Windows, version 28 (IBM, Chicago, IL, USA).

The plasma and hepatic biochemical parameters, fat in feces, hepatic gene expression, thickness of fat streaks in the aorta arch, and main FA molar concentration were selected, log-transformed when necessary, and scaled with the unit variance (UV); then, a heatmap was created by using hierarchical clustering analysis (HCA) in RStudio version 1.3.959 and R version 3.6.3 (San Francisco, CA, USA) (R Core Team, 2020) with the “gplots” package [20]. The aim was to analyze the clustering of parameters and hamster groups depending on their metabolic responses to different diets.

## 3. Results

### 3.1. Hamsters’ General Performance and Plasma Parameters

The performance of the hamsters fed the experimental diets based on olive oil (OO7 and OO21) and milk fat (MF7 and MF21) was similar (Table 2). The energy intake was higher in the high-fat-diet-fed animals. On the other hand, the fat content of the feces changed with the fat content. The OO21 diet caused a higher rate of fat excretion than the other diets. This might account for the lower weight gain, especially for the animals in the OO21 group, although there was no statistical significance, which was probably due to the great variability.

Both the type and percentage of fat in the diet had a significant effect on most of the plasma biochemical parameters in the hamsters (Figure 1). Moreover, there was a significant interaction between both factors for all parameters, indicating that the effect of increasing the fat content was different for each type of fat. Thus, the high-fat diet caused an increase in TC, CE, and total FA concentrations only in hamsters fed the MF21 diet. The concentrations of TG and glucose were higher in both high-fat-fed animal groups, although the increases were greater in the hamsters that were fed MF.

Finally, although the method used allowed for the analysis of various tocopherols and retinoids, only α-tocopherol and retinol were detected in the plasma of the hamsters. The high-fat diets caused an increase in the retinol concentration, regardless of the type of fat, while the concentration of α-tocopherol doubled only in hamsters that were fed the MF21 diet. 

The details of the plasma FA composition (molar concentration) are shown in Appendix A. The FA profiles of the plasma, in molar percentages, were very similar in both groups that were fed a normal-fat diet, regardless of the type of fat ingested, as we previously described [21]. Figure 2 shows the plasma FA in which the molar proportion changed when comparing hamsters fed a normal-fat diet and a high-fat diet. More than 40 FAs were detected in the hamsters’ plasma, but among them, only the proportions of 4 (OA, stearic acid, linoleic acid, and docosahexaenoic acid (DHA)) significantly changed when the OO was increased from 7% to 21% in the hamsters’ diets. These FAs were found in high proportions in OO, except for DHA, which was not detected in the oil (Appendix A). The molar concentrations of most FAs increased in the MF21 group in comparison with those in the MF7 group. However, only the molar proportions of five FAs (VA, RA, α-linolenic acid, eicosapentaenoic acid (EPA), and DHA) changed with increasing dietary fat content. These FAs were in relatively low proportions both in MF and in the hamsters’ plasma. DHA was the only FA whose proportion increased in animals fed a high-fat diet with the two types of fat. In addition, there were no significant differences in the proportions of DHA between the OO7 and MF7 groups or between the OO21 and MF21 groups.

### 3.2. Hamsters’ Hepatic Lipids

No statistical differences were detected in total fat in the liver between hamster groups (Table 2). TC and CE concentrations in the livers of MF7-fed hamsters were lower than those of other groups (Figure 3). On the other hand, the TG concentration decreased significantly only in the hamsters fed the OO21 diet compared with OO7-fed animals.

### 3.3. Hepatic Gene Expression

The variability found in the relative expression of the hepatic genes analyzed was large, which made it difficult to detect significant differences between the groups (Table 3). Despite this, it was observed that the type of fat significantly affected the expression of the gene related to reverse cholesterol transport, SR-B1; the LDLR gene responsible for the clearance of LDL from the circulation; and the HMG-CoAR and ACAT genes, which encode the enzymes HMG-CoA reductase and acyl-CoA: cholesterol acyltransferase, which in turn regulate cholesterol synthesis and catalyze CE synthesis in the liver, respectively. All of these genes were upregulated in the OO-fed hamsters. The percentage of fat in the diets influenced the expression of the FA transporter CD36 gene and that of the ACAT gene. The expression of both genes was slightly higher in the hamsters fed normal-fat diets.

### 3.4. Aortic Histology

Figure 4A–D shows representative examples of the fat-streaked lesions stained with Oil Red O in the aortic arch of the hamsters of different groups. A quantitative assessment of lesion size showed that the thickness of the fatty streaks depended on the fat content of the diet, but not on the type of fat (Figure 5). Oil Red O dye was also found to accumulate in the adventitia (Figure 4E,F) in the high-fat-fed hamsters, which is also related to the development of atherosclerosis [22].

### 3.5. Clustering Analysis

Taking into account all of the analyzed parameters, the OO-fed hamsters seemed to be the animals that were most similar to each other because they were grouped together (Figure 6), while hamsters fed the MF21 diet were shown to be the most segregated group because of their differentiated characteristics. Furthermore, MF7 was clustered closer to the OO-fed hamsters than the MF21-diet-fed ones, indicating that the MF21 diet caused the greatest changes in the animals’ characteristics.

By comparing the OO7 and MF7 groups, it could be seen that the principal difference between the animals was the higher expression of most hepatic genes (CD3, LXR, PPARa, ApoB, ACAT, SR-B1, LDLR, and HMGCoAR) in the OO7-diet-fed hamsters, while the hamsters in the OO21 group were characterized by the expression of only genes related to reverse cholesterol transport (SR-B1 and LDLR). On the other hand, the MF21 group was clearly characterized by the high concentration of most plasma parameters (most FAs, cholesterol fractions, TGs, and α-tocopherol). Finally, the high-fat-diet-fed animals shared a high concentration of plasma TGs, OA, stearic acid, plasma glucose, and DHA, as well as a higher value of aortic fat streak thickness. 

Regarding the clustering of the parameters, with some exceptions, the plasma parameters and hepatic parameters appeared to be segregated in different clusters. The hepatic parameters formed two clusters. One related hepatic TGs to most genes related to lipid metabolism and lipoprotein synthesis, while the other related hepatic TC and CE to the expression of genes related to the reverse transport of cholesterol. This cluster included the gene expression of HMGCoAR, fat in the liver, and fat in feces. On the other hand, it is worth mentioning that the parameter related to the progression of atherosclerosis (AFST) appeared to be grouped with the plasma concentration of TGs, glucose, and DHA but not with the plasma cholesterol fractions.

## 4. Discussion

In 2010, Dillard et al. [11] reviewed the scientific works that had been published at that time on the relationship among diet, plasma lipid fractions, and the development of atherosclerosis in hamsters. This review included many works in which hamsters were fed butter or olive oil in different proportions. However, to the best of our knowledge, so far, no studies have been published that compare the different effects of these two fats, which are habitual components of the human diet, in two different proportions in the same study, which is of great interest given the variable responsiveness among different hamster strains and studies [11,23]. The hamster model used in our study was previously found to be appropriate for studying atherogenesis in association with dietary changes [24,25]. Moreover, cholesterol was added to the OO diets to obtain the same cholesterol content that MF-based diets naturally have. Thus, the amount of cholesterol in the high-fat diets (0.047%) was lower than what has commonly been used in hamster studies (from 0.05 to 3%) [11,25]. These latter levels exceed what is considered high in humans [11]. Therefore, we chose an amount of cholesterol that simulated physiological conditions, thus avoiding mixing up the effect of the FA profile of each fat with the effect of cholesterol.

The principles that dictate the effects of dietary lipids on plasma lipoprotein, TG, and cholesterol levels have been known for a long time [26] because of numerous experimental studies on animals and humans. These studies have elucidated that the concentration of LDL-C usually increases as the level of dietary cholesterol increases. In addition, in general, the consumption of TGs with predominant SFA content further increases the concentration of cholesterol carried in this lipoprotein fraction, while those containing predominantly unsaturated FAs lower these levels [27,28]. These general principles were fulfilled in the present study. 

In the same way, it is well established that the plasma levels of cholesterol and TGs are highly conditioned by hepatic lipid metabolism. The lipid content in the liver reflects the balance between two principal processes: the incorporation of lipids (both coming from the diet and produced in the liver) into VLDL particles and their secretion into the plasma, as well as the removal of plasma lipoproteins via receptor-mediated endocytosis. 

The primary stimulus for VLDL formation and secretion is the availability of FAs for TG synthesis. Consequently, an increase in plasma TGs was observed in the hamsters fed high-fat diets. Nevertheless, the changes detected in our study were less pronounced than those found in other studies [10,24,26], which was most likely due to the low cholesterol content in the hamsters’ diets in comparison with those in other studies. It has been demonstrated that cholesterol-enriched diets increase VLDL production and secretion [29,30]. Moreover, the type of fat also affects these processes. Sessions et al. [27] found higher VLDL production in hamsters that were fed lard (which is rich in saturated triglycerides) than in those that were fed OO. This is in agreement with the higher plasma TG concentration observed in the MF21-fed hamsters than in the OO21-fed ones.

On the other hand, high-fat diets caused an increase in the TC and CE concentrations in the liver of MF21-fed hamsters that was reflected in the plasma cholesterol fractions. According to Dietsy et al. [31], when excess cholesterol is delivered to the liver, there is a partial suppression of LDLR gene expression activity and an increase in hepatic CE. They explained that LDLR gene expression in the liver is regulated by the sterol content of the cell. When the content of cell sterol is low, the transcription of the LDLR gene is activated and vice versa. According to the hypotheses of some authors [26,32,33], the fraction of cellular cholesterol involved in the sterol-mediated regulation of LDLR gene expression is likely to be a very small pool of unesterified cholesterol (CR). CR is in equilibrium with a much larger storage pool of CE. Nevertheless, dietary FA may alter the distribution of cholesterol within the hepatocytes between CR and CE fractions [33]. The reason for this is the activity of the cholesterol-esterifying enzyme ACAT. The activity of ACAT is known to be higher in animals fed unsaturated FAs than in animals fed SFAs. Thus, SFAs may inhibit ACAT activity, resulting in an expansion of the putative CR pool and a reduction in the LDLR mRNA level and activity [32]. Conversely, unsaturated FAs, such as OA, which is the preferred substrate for ACAT, increase the production of CE in the liver, reduce the CR pool, and increase LDLR activity [31]. Changes in the ACAT mRNA level were small among animals fed different diets in the present study. Thus, the effect on LDLR expression seemed to be more related to substrate availability and the preference for ACAT. Other studies found differences in hepatic ACAT activity without changes in mRNA levels [34]. In the present study, the effect of cholesterol on LDLR expression could not be appreciated because the increase in the cholesterol content in the diet (from 0.015 to 0.047%) was small. Daumiere et al. [32] observed an increase of 25% in LDLR activity when the cholesterol content of the diet increased from 0 to 0.06%. However, the effect of the type of fat on LDLR gene expression was clearly seen in our study and explained the higher CE concentration found in the OO7-diet-fed hamsters than in the MF7-fed animals. However, the liver CE concentration did not increase in the OO21 group. It has been speculated that an increase in LDLR activity and redistribution of intracellular cholesterol toward CE might also increase cellular HMG-CoA reductase activity [35], which was in agreement with the higher HMG-CoAR gene expression observed in the OO-fed hamsters. The accumulation of hepatic CE in the OO21-fed hamsters could increase biliary cholesterol secretion [36], leading to supersaturated bile, which could, in turn, explain the high proportion of fat in feces observed in this group (Table 2); this is in agreement with the cluster analysis, where these two parameters appeared to be close to each other.

In summary, the increase in VLDL production and secretion due to high-fat diets contributed to the increase in plasma LDL-C associated with the increase in cholesterol in the diets. However, OO attenuated this effect because it caused the upregulation of the expression of the LDLR and SR-B1 genes. Hepatic LDLR and SR-B1 are two major membrane receptors responsible for the uptake of LDL and HDL from plasma [37]. The clustering analysis confirmed the association between OO consumption and the upregulation of genes related to lipid metabolism and reverse cholesterol transport.

The differential changes in plasma lipid fractions among the hamster groups were reflected in the plasma FA profiles, as has been previously observed in hamsters [21]. The OO21 diet only caused an increase in TGs, which was reflected in an increase in the proportions of OA and stearic acid, which are the main contents of these types of lipids [9], while linoleic acid is the main FA in CE, whose concentration decreased in the plasma of the OO21-diet-fed hamsters in relation to total FAs. On the other hand, the MF21-diet caused an increase in all lipid fractions, which was reflected in the increase in the total FA concentration, but only the proportion of minor FAs increased. RA and VA have been previously described as dairy fat consumption markers in humans [38]. Burdge et al. [39] demonstrated that, when RA was consumed in a diet, it was incorporated in all lipid fractions, but preferentially in CE, while VA was preferentially incorporated in TGs and could not be found in CE. This fact can partially explain the higher concentration of RA than that of VA in plasma (the opposite occurred in MF; Appendix A), as the CE concentration in the plasma was higher than that of the TGs (Table 2). In addition, the endogenous desaturation of VA into RA due to stearoyl-CoA desaturase-1 (SCD1) activity in hamsters was previously observed [40].

The SCD1 activity indices, which are expressed as the product-to-substrate ratio in the plasma, are shown in Appendix A. The *cis*9-18:1/18 index did not change, regardless of the type or fat content in the diet. Miller and Ntambi [41] highlighted the importance of keeping the ratio of OA to stearic acid constant to maintain membrane fluidity. Small changes in this ratio can affect the ability of cells to respond to external stimuli. Because of that, it is a highly regulated ratio in humans, as it also seems to be in hamsters. The other SCD1 indices were higher in the MF-fed animals than in the OO-fed animals. The reason may be the fact that SFAs are strong activators of SCD1, which is a defense mechanism against the negative effects of SFAs in cells, as speculated by Mauvoisin and Mounier [42]. This high SCD1 activity may account for the almost tenfold increase (Appendix A) in plasma RA concentration in the MF21-diet-fed hamsters. Santora, Palmquist, and Roehrig [43] reported that VA taken with food can increase the concentration of RA in rats by 6–10 times, which also occurred in the present study.

On the other hand, the increase in the EPA proportion in the plasma of hamsters that were fed milk fat was consistent with the higher concentration of α-linolenic acid (the precursor of longer PUFA n-3) in their diets [44]. More difficult to explain is the fact that the proportion of plasma DHA increased in the same proportion in the two high-fat diets. Wien, Rajaram, Oda, and Sabaté [45] mentioned that the limiting step in the metabolism of the PUFA n-3 family is the saturation of the membrane with DHA. This argument can explain why differences were found in the individual n-6 and n-3 FAs in the hamsters, but the DHA percentage stayed constant among the hamsters that were fed diets with the same fat content, regardless of the type of fat. The increased proportion of DHA in the plasma of the high-fat-diet-fed animals might be related to the fact that this FA is incorporated into TGs more than it is into CE [9]. Both the DHA proportion and TG concentration increased approximately twofold when comparing the normal-fat- and high-fat-fed animals, and they appeared to be very closely related in the clustering analysis.

High-fat diets caused an increase in the thickness of fat streaks in the aortic arches of hamsters in comparison with the normal-fat-fed animals, regardless of the type of fat. One of the surprising findings of this work is that, although the plasma lipid profile of the MF21 group can be described as more atherogenic than that of the OO21 group, because of the higher TC and TG concentrations [46], the progression of atherosclerosis seemed to be similar in both animal groups. The development of fat streaks in the high-fat-fed animals seemed to be more closely related to the plasma TG and glucose concentrations (as shown in the clustering analysis). It has long been accepted that atherosclerosis is a multifactorial disease and that high-fat diets, which cause hypertriglyceridemia and hyperglycemia, may contribute to its development [12]. Nevertheless, there were some studies that found that OO was associated with a healthier plasma lipid profile than that of butter [6,47], but, to the best of our knowledge, there are no published studies comparing the effects of MF and OO on the progression of atherosclerotic lesions in the same hamster strain. There were some studies that compared OO with other fats that are rich in SFAs, such as coconut oil [46,48]. In the study of Wilson et al. [48] it was observed that, in the OO-fed hamsters, CE deposition in the aortic wall was 60% lower than in coconut-oil-fed hamsters. Mangiapane et al. [46], on the other hand, showed that OO caused a greater regression of atherosclerotic lesions in the aortic arch than coconut oil did in hamsters that were previously maintained on an atherogenic diet. Thus, our result would suggest that the detrimental effect of SFAs in MF on atherogenesis could be modulated by compounds that are present in it that can counteract their negative effects. Martin et al. [25] suggested that variations in macro- and micronutrient contents can drastically influence the atherogenic capacity of dairy fats. They also found that fasting plasma TC was not the best predictive factor for atherosclerosis in hyperlipidemic hamsters. Moreover, in their study, the expression levels of hepatic genes related to lipid metabolism and transport were the main quantitative contributors to the atherogenic index. Their analysis gave SCD1 gene expression a determining role in the atherogenic score of each fat. The authors related the upregulation of SCD1 to increased endogenous lipogenesis, and more severe atherosclerosis was observed when hamsters were fed raw milk butter. However, in our study, the SCD1 activity seemed to be related to high RA production. We previously speculated that the consumption of MF, which is rich in SFAs, increases SCD1 activity, which, in turn, causes a great increase in the plasma RA content. Therefore, RA might be responsible for the fact that the MF21-fed hamsters did not develop more lesions than the OO21-fed animals, despite their disadvantageous lipid profile.

There are several studies that have demonstrated that RA reduces atherosclerotic plaque formation in hamsters. In published works, RA was administered to hamsters as a natural component of dairy fat, or, in some cases, the dairy fat was enriched with RA through different means [24,40,49]. In other cases, RA was added—purified or as a CLA mixture—to diets containing fats from different sources [17,50,51,52,53,54]. Thus, when evaluating the effects of RA on atherosclerosis, there are many variables that have to be considered—among others, the hamster strain [11,23], the dietary fat and cholesterol contents, the FA profile, the proportion at which CLA is administered, and the composition [40]. In all of these respects, there is great variability in the published works; therefore, the results are, in some cases, inconsistent. 

In general, the beneficial effects of CLA have been related to two different aspects: the improvement of the atherogenic profile of plasma lipoproteins (decreases in non-HDL-C and nonHDL-C:HDL-C ratio) and/or the direct effect of CLA on the inflammatory process that occurs during the formation of atheroma plaque. From the analysis of the published studies, it can be concluded that, when administered in natural proportions as part of an equilibrated diet (in terms of fat and cholesterol contents) [50,54], RA has no effect on plasma lipid fractions, which is in agreement with our results when comparing the MF7 and OO7 hamster groups.

On the other hand, when RA was given to hamsters as part of a hypercholesterolemic diet (high-fat and/or high-cholesterol) [17,24,40,49,51], the results for the plasma parameters were inconsistent, and the effects were difficult to separate from the effects caused by the fat and cholesterol contents of the diets. However, a common observation across all studies is that of a lower progression or regression of aortic atherosclerotic lesions seen in RA-fed hamsters compared with non-RA-fed hamsters.

For all of these reasons, it could be concluded that RA’s ability to protect against the progression of atherosclerotic lesions seems to be due to a direct effect on the inflammatory process that occurs during the formation of atheroma plaque. Wilson et al. [55] speculated that the slowing of atherosclerosis by RA is due to changes in the oxidative susceptibility of LDL particles. Other studies also found that the degree of early aortic atherosclerosis in hypercholesterolemic hamsters fed OO [52] or other animal- and plant−derived fats [56] was correlated with the oxidative susceptibility of lipoprotein and not with the cholesterol levels of lipoprotein.

Antioxidant protection has been associated, in some cases, with the α-tocopherol-sparing capacity of CLA [52], as well as that of OO [7], in hamsters. We could not measure the α-tocopherol content of LDL, but the plasma concentration generally reflected the LDL-α-tocopherol concentration [57]. In the current study, the highest plasma α-tocopherol concentration corresponded to the MF21-diet-fed hamsters (79.36 nmol/L, 34.3 µg/L; Figure 1), but the α-tocopherol/TC ratios were similar in all animal groups. Parker et al. [58] found that the half-maximal inhibition of atherogenesis in hamsters was associated with approximately 20 µg of α-tocopherol per milliliter of plasma. Therefore, potential oxidation resistance did not seem to be conferred by α-tocopherol in the present study.

Bruen et al. [58] reviewed the processes that take place in the development of atherosclerotic lesions and the literature dealing with the effects of CLA in this process. As they explained, the development of atherosclerotic lesions is initiated by endothelial dysfunction at arterial points that facilitate the passage and retention of macromolecules, such as LDL molecules, within the intima layer. The subsequent oxidation of LDL within the subendothelium caused by reactive oxygen species (ROS) triggers an inflammatory response. The majority of leukocytes within the developing atherosclerotic lesion are monocytes and macrophages. In their review, they concluded that monocytes/macrophages are the cellular targets through which RA mediates its effects. RA inhibits monocyte/macrophage adhesion and migration, foam cell formation, and the generation of inflammatory mediators via PPAR-γ- and LXRα-dependent mechanisms. Thus, Valeille et al. [49] observed that RA, which is a ligand with high affinity and an activator of the PPAR α and γ transcription factors, causes the downregulation of the expression of pro-inflammatory genes (IL-1β, COX-2, and TNF-α), whose products influence ROS generation [59]. 

In summary, our results indicate the anti-atherosclerotic effect of RA is not related to its effects on the plasma lipid profile, nor to its α-tocopherol-sparing effect. Thus, our results support the hypothesis that RA, when ingested in amounts that are naturally present in MF, protects against LDL oxidation and the progression of atherosclerosis by regulating the expression of genes that influence ROS generation. 

The limitations of this study include the following: (1) Because of the small blood volume obtained from each hamster, it was not possible to analyze lipoprotein fractions or their oxidizability. (2) The variability in the values obtained for some parameters (for example, the expression of some genes) was very high, making it difficult to determine significant differences and to interpret the data. (3) It would have been interesting to include two more groups to which we would have given MF without VA and RA in the design. However, this would have greatly increased the number of animals required, along with the ethical considerations that this would entail. (4) As the potential beneficial effect of RA was seen in the MF21 diet, the results are difficult to extrapolate to humans, since a high-fat diet based only on MF is not realistic in humans. 

## 5. Conclusions

Using the Golden Syrian hamster as an animal model, this research demonstrated that MF rich in VA and RA had a similar effect to that of OO on the progression of atherosclerosis if it was ingested in a similar proportion and with the same amount of cholesterol. 

The type of fat did not affect most plasma parameters when hamsters were fed normal fat diets. However, the OO diets caused the upregulation of LDLR and SR-B1 gene expression in the liver, and, consequently, a higher hepatic CE concentration was found in hamsters fed the OO7 diet than in animals fed the MF7 diet.

High proportions of dietary fat increased atherosclerosis-associated lesions regardless of the origin of the fat, which suggests that RA is effective in balancing the negative effects of the high SFA content in milk fat, although MF further modified the values of certain indicators that are usually associated with atherosclerosis (i.e., TG and CE in plasma).

In perspective, the generally attributed pro-atherogenic nature of MF should be reconsidered and related to the presence of anti-atherogenic compounds, such as RA, whose concentration can be augmented using natural or technological means. 

## Figures and Tables

**Figure 1 metabolites-13-00827-f001:**
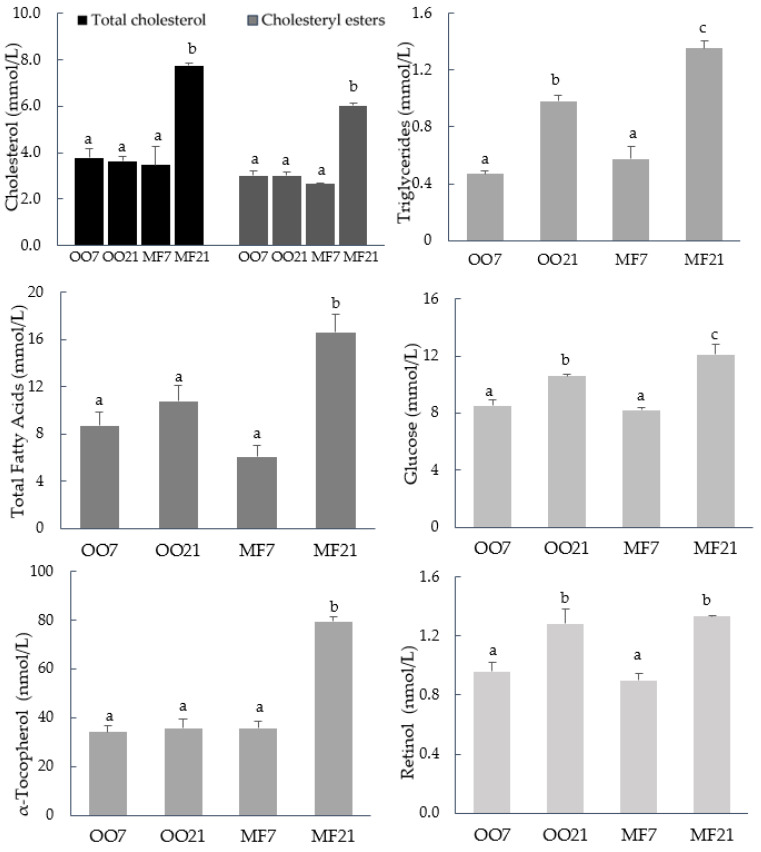
Plasma biochemical parameters of hamsters fed a diet containing 7% (*w*) olive oil (OO7), fed a diet containing 21% (*w*) olive oil (OO21), fed a diet containing 7% (*w*) milk fat (MF7) and fed a diet containing 21% (*w*) milk fat (MF21). Bars represent mean + SD. Bars with different letters in the same group show a significant difference at *p* ≤ 0.05.

**Figure 2 metabolites-13-00827-f002:**
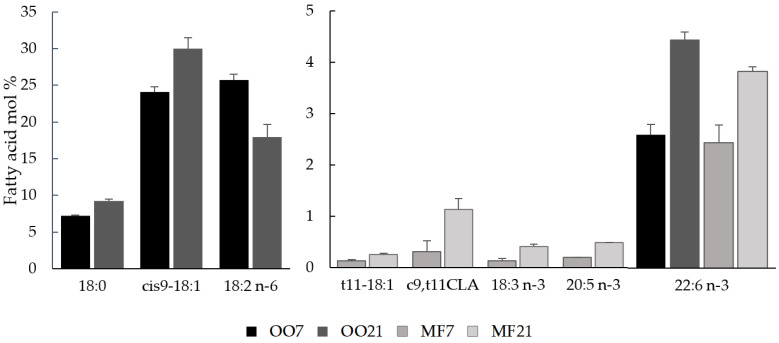
Plasma fatty acids that were in different proportions (mol%) in hamsters that were fed normal-fat and high-fat diets. OO7 hamsters were fed a diet containing 7% (*w*/*w*) olive oil, OO21 hamsters were fed a diet containing 21% (*w*/*w*) olive oil, MF7 hamsters were fed a diet containing 7% (*w*/*w*) milk fat, and MF21 hamsters were fed a diet containing 21% (*w*/*w*) milk fat. CLA—conjugated linoleic acid.

**Figure 3 metabolites-13-00827-f003:**
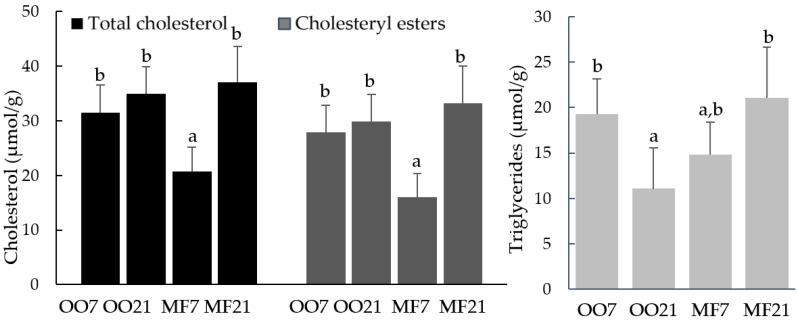
Lipid fractions in the livers of hamsters fed a diet containing 7% (*w*) olive oil (OO7), fed a diet containing 21% (*w*) olive oil (OO21), fed a diet containing 7% (*w*) milk fat (MF7), and fed a diet containing 21% (*w*) milk fat (MF21). Bars represent mean + SD. Bars with different letters in the same group show a significant difference at *p* ≤ 0.05.

**Figure 4 metabolites-13-00827-f004:**
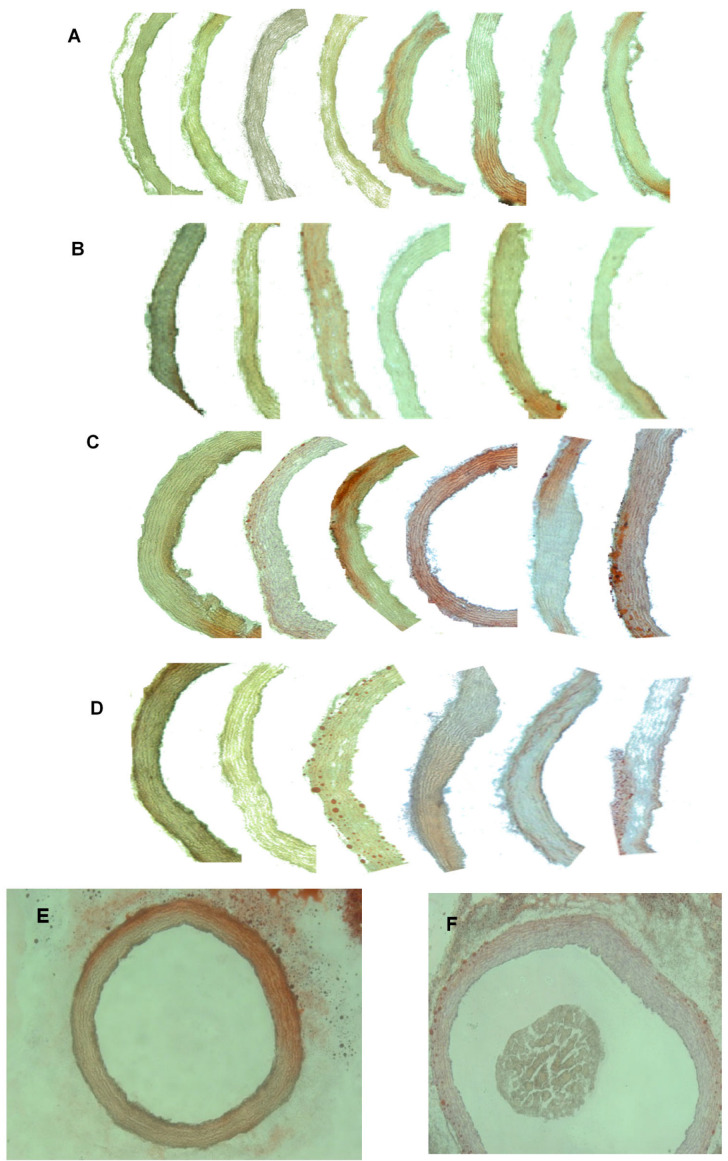
Hamsters’ aortic arch sections showing fatty streaks stained with Oil Red O (×10): (**A**) hamsters fed a diet containing 7% olive oil; (**B**) hamsters fed a diet containing 7% milk fat; (**C**) hamsters fed a diet containing 21% olive oil; (**D**) hamsters fed a diet containing 21% milk fat. Fat accumulation can be seen in the adventitia of hamsters that were fed a diet containing 21% olive oil (**E**) and hamsters fed a diet containing 21% milk fat (**F**).

**Figure 5 metabolites-13-00827-f005:**
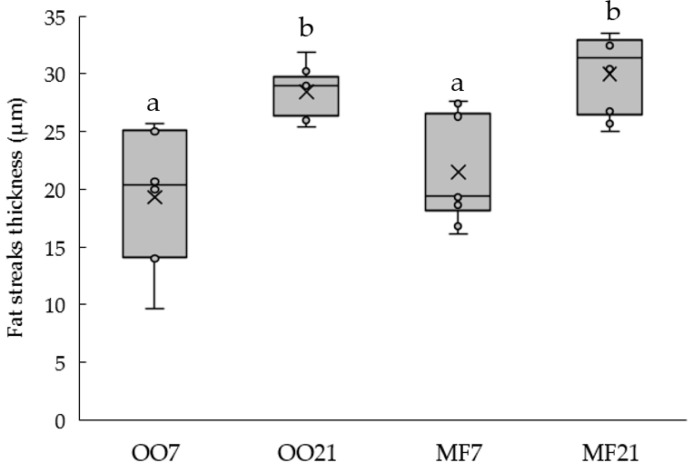
Fat streak thickness distribution in hamsters’ aortic arches. OO7, hamsters fed a diet containing 7% olive oil. OO21, hamsters fed a diet containing 21% olive oil. MF7, hamsters fed a diet containing 7% milk fat. MF21, hamsters fed a diet containing 21% milk fat. The midline represents the median, the “×” represents the mean value, box borders represent the 25 and 75% of values, and whiskers represent the minimum and maximum values. The letters a and b indicate significant differences (*p* ≤ 0.05) in the compared groups.

**Figure 6 metabolites-13-00827-f006:**
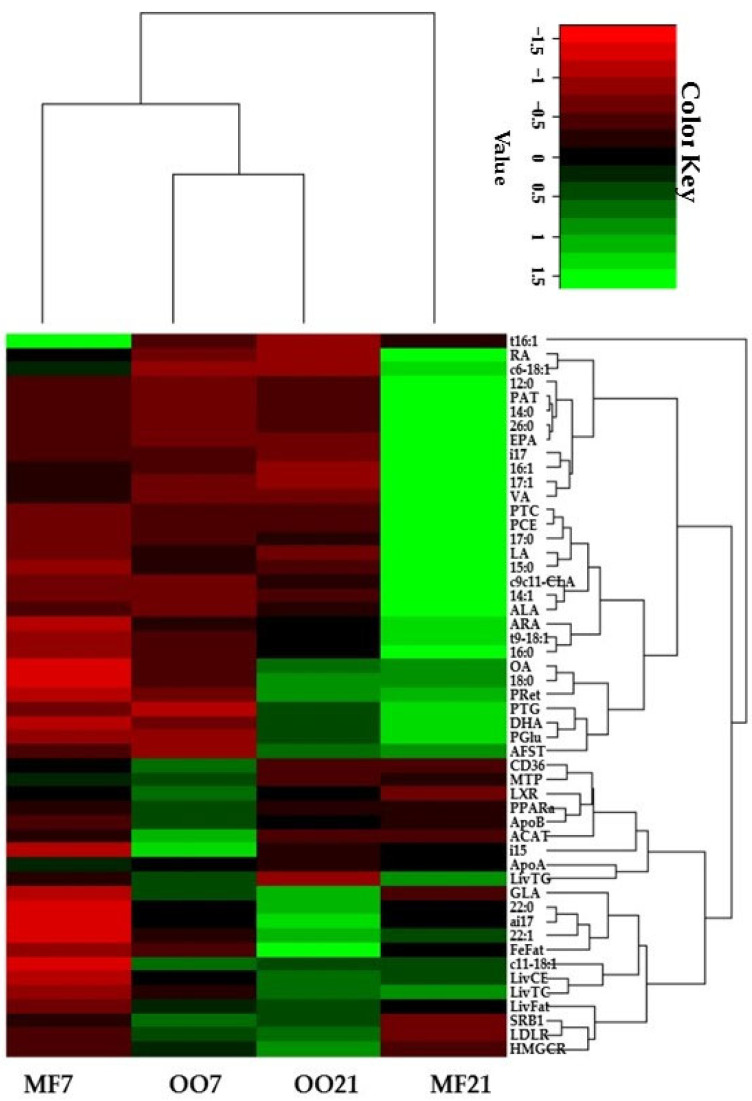
Heatmap of the hierarchical clustering analysis. OO7, hamsters fed a diet containing 7% olive oil. MF7, hamsters fed a diet containing 7% milk fat. OO21, hamsters fed a diet containing 21% olive oil. MF21, hamsters fed a diet containing 21% milk fat. AFST: aortic fat streak thickness; ai17: *anteiso*17:0ALA: α−linolenic acid; ARA: arachidonic acid; c6−18:1: *cis6−*18:1; c9c11−CLA: *cis*9,*cis*11−conjugated linoleic acid; DHA: docosahexaenoic acid; EPA: eicosapentaenoic acid; FeFat: fat in feces; GLA: γ −linolenic acid; i15: *iso*15:0; i17: *iso*17:0; LA: linoleic acid; LivCE: liver cholesteryl ester; LivFat: fat in liver; LivTC: total liver cholesterol; LivTG: liver triglycerides; OA: oleic acid; PAT: plasma α−tocopherol; PCE: plasma cholesteryl ester; PGlu: plasma glucose; PRet: plasma retinol; PTC: total plasma cholesterol; PTG: plasma triglycerides; RA: rumenic acid; t16:1: *trans*9−16:1; t9−18:1: *trans*9−18:1; VA: vaccenic acid. Expression of hepatic genes: ACAT: acyl−CoA: cholesterol acyltransferase; ApoA: ApoA1; ApoB: ApoB100; CD36: cluster of differentiation 36; HMG−CoAR: hydroximethylglutaryl−CoA reductase; LDLR: LDL receptor; LXR: liver X receptor; MTP: microsomal triglyceride transfer protein; PPARa: peroxisome proliferator−activated receptor α; SRB1: HDL scavenger receptor B1.

**Table 1 metabolites-13-00827-t001:** Macro- and micronutrient compositions of the hamster diets.

	Hamster Group			
Component	OO7 ^1^	OO21 ^2^	MF7 ^3^	MF21 ^4^
	g/kg			
Protein	180.8	216.7	180.6	216.5
Carbohydrates	511.5	357.4	511.7	356.0
Fat	70.0	208.3	70.4	208.5
Fiber	64.6	60.5	64.6	60.5
Mineral mixture ^5^	41.4	52.75	41.4	52.75
Vitamin mixture ^5^	10	10	10	10
Cholesterol ^6^	0.155	0.471	0.157	0.472
Energy (kcal/g)	3.4	4.2	3.4	4.2
	% Energy			
Protein	21.2	20.8	21.3	20.8
Carbohydrates	60.1	34.3	60.2	34.2
Fat	18.6	45	18.5	45

^1^ Diet containing 7% (*w*/*w*) olive oil. ^2^ Diet containing 21% (*w*/*w*) olive oil. ^3^ Diet containing 7% (*w*/*w*) milk fat. ^4^ Diet containing 21% (*w*/*w*) milk fat. ^5^ Mineral and vitamin mixtures prepared by Harland Laboratories (Madison, WI, USA) for Teklad diets. ^6^ Present naturally in diets based on milk fat and added to diets based on olive oil to obtain the same concentration of cholesterol for each fat content.

**Table 2 metabolites-13-00827-t002:** General performance parameters of the hamsters (means (SD)).

Hamster Groups						
	OO7 ^1^	OO21 ^2^	MF7 ^3^	MF21 ^4^	*p* ^5^	*p* ^6^
*General performance*						
Initial weight (g)	87.5 (8.4)	93.7 (8.2)	90.7 (7.4)	88.2 (3.6)	0.527	0.722
Final weight (g)	117.5 (17.3)	117.2 (23.1)	121.0 (12.2)	122.3 (8.4)	0.698	0.788
Weight gain (g)	32.5 (12.4)	24.2 (14.8)	33.3 (8.72)	40.9 (6.33)	0.264	0.933
Food intake (g/d)	7.49 (0.685)	7.45 (0.422)	7.45 (0.630)	7.06 (0.476)	0.258	0.308
Energy intake (kcal/d)	26.0 (1.74)	31.3 (1.77)	25.3 (2.14)	29.6 (2.00)	0.497	<0.001
Fat in feces (g/100 g)	1.67 (0.0423)	2.54 (0.210)	1.46 (0.234)	1.85 (0.206)	0.248	0.021
Fat in liver (g/100 g)	6.90 (1.01)	7.17 (1.98)	5.56 (0.83)	6.58 (1.61)	0.065	0.306

^1^ Hamsters fed a diet containing 7% (*w*/*w*) olive oil. ^2^ Hamsters fed a diet containing 21% (*w*/*w*) olive oil. ^3^ Hamsters fed a diet containing 7% (*w*/*w*) milk fat. ^4^ Hamsters fed a diet containing 21% (*w*/*w*) milk fat. ^5^ *p*-value of the effect of the type of fat. ^6^
*p*-value of the effect of the fat percentage. *p* ≤ 0.05 was considered statistically significant.

**Table 3 metabolites-13-00827-t003:** Relative expression (mean (SD)) of hepatic genes in hamsters fed different diets. Reference gene: β-actin.

Gene	OO7 ^1^	OO21 ^2^	MF7 ^3^	MF21 ^4^	*p* ^5^	*p* ^6^
CD-36	0.7967 (0.3748)	0.5054 (0.1966)	0.5933 (0.1163)	0.5024 (0.1302)	0.719	0.021
PPAR-α	2.742 (0.6497)	2.294 (0.4988)	2.268 (0.8301)	2.156 (0.7124)	0.090	0.736
SR-B1	2.985 (0.6695)	2.536 (0.9361)	1.899 (0.4067)	1.770 (0.908)	0.012	0.312
ACAT	0.3704 (0.04081)	0.2825 (0.06138)	0.2924 (0.04305)	0.2827 (0.04706)	0.040	0.036
ApoB100	3.387 (0.5567)	2.988 (0.4073)	2.570 (0.8832)	2.774 (0.8040)	0.341	0.662
ApoA1	0.4890 (0.08334)	0.4319 (0.04819)	0.5134 (0.2079)	0.4949 (0.2400)	0.382	0.392
LDLR	0.7553 (0.1160)	0.7059 (0.1766)	0.4572 (0. 09785)	0.4243 (0.2668)	0.005	0.979
HMG-CoAR	0.5781 (0.1295)	1.220 (1.072)	0.4823 (0.1212)	0.3833 (0.1174)	0.024	0.856
MTP	1.488 (0.5130)	1.013 (0.2243)	1.359 (0.7599)	1.079 (0.3730)	0.607	0.052
LXR-α	0.8654 (0.3617)	0.6443 (0.1251)	0.7078 (0.3158)	0.5249 (0.1664)	0.057	0.082

^1^ Hamsters fed a diet containing 7% (w) olive oil. ^2^ Hamsters fed a diet containing 21% (w) olive oil. ^3^ Hamsters fed a diet containing 7% (w) milk fat. ^4^ Hamsters fed a diet containing 21% (w) milk fat. ^5^ *p*-value of the effect of the type of fat. ^6^ *p*-value of the effect of the proportion of fat. *p* ≤ 0.05 was considered statistically significant.

## Data Availability

Data is contained within the article or Appendix A.

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
