# Peer review of "Normal-Fat vs. High-Fat Diets and Olive Oil vs. CLA-Rich Dairy Fat: A Comparative Study of Their Effects on Atherosclerosis in Male Golden Syrian Hamsters"

_metabolites, 2023, doi:10.3390/metabo13070827_

Round 1

Reviewer 1 Report

GENERAL:

The paper of Berriozabalgoitia et al. examines the effect of several types of diets on atherosclerosis in hamsters.

Although the paper is interesting and falls into scope of the Journal, the work is difficult to understand, mainly because of poor grammar and many typos. Extensive English corrections are strongly recommended. The abstract should be written in such a way that the reader can understand it well, while there are several confusions in this abstract due to poorly constructed sentences. I strongly recommend that the authors use formal scientific language rather than simple words and phrases such as "notoriously."

SPECIFIC:

TITLE:

Avoid two sentences in the title! The title is too general and more detail is needed, such as the hamster breed (for example: male golden syrian hamster)

ABSTRACT

The abstract, which is a summary of the entire paper, is quite poorly written, incomprehensible, with sentences that are too long and poorly constructed. Please consider the possibility of proofreading by a native English speaker.

INTRODUCTION:

the introduction is difficult to understand - although I understand the context of the study to some extent, the sentences are rather confused and poorly constructed grammatically.

Lines 69-77 should be restructured to make the structure of the study more understandable. For example, phrases such as "very different FA content" should be avoided, as this is not an acceptable scientific definition term.

MM

Experimental design. Since there are 4 groups (OO7, OO21, MF7, MF21), it is not necessary to list further definitions as this will confuse the reader about the exact number of groups, i.e. lines 90, 91: normal fat diet, NFD, OO7 and MF7, and 21% - high fat diets, HFD, OO21 90 and MF21. After these sentences, the reader is not sure about how many groups are ultimately involved in the experiment.

Section 2.4. the method should be written in more detail, because the method description is for other scientists to repeat the procedure given in the description.

RESULTS

For Table 2, I suggest that autohrs present the significantly changed variables in graphs rather than in a table because the visibility of differences is greater with a graphical presentation. Other non-significant results can be presented in the table.

The authors note that "there is a significant interaction between the two factors for all parameters (data not shown), suggesting that the effect of increasing fat content is different for each type of fat." Lines 225-226. I think it is important to define exactly for which type of fat the interaction between type of fat and percent fat is important. Please define it in the results along with the interaction effect in the graph/table for each dependent variable.

Table 4. as with table 2, I suggest that autohrs present significantly changed variables in graphs rather than tables because the visibility of differences is greater with a graphical representation. Other non-significant results can be presented in the table.

DISCUSSION

The discussion is much too long. The authors should shorten it to a maximum of three pages. Please do not extensively describe the work of others to discuss your findings. One or two sentences with citations of other authors and one sentence with your opinion of that particular result are sufficient.

Did the authors explain/discuss the result of the cluster analysis (lines 346-349)? How to explain the observed result when clustering the parameter AFST and plasma concentrations of TG, glucose, and DHA but not plasma cholesterol fractions?

Author Response

Thank you very much for reading the manuscript, for your work and for making comments and suggestions to improve the work. We really appreciate it.

As suggested, the manucript has been sent to Mdpi’s English Services and all proposed changes has been incorporated.

Specific comments:

TITLE:

Comment “Avoid two sentences in the title! The title is too general and more detail is needed, such as the hamster breed (for example: male golden syrian hamster).

Response:

The title has been changed as proposed by the English editor and, in addition, we included the hamsters’ breed in the title. Thank you very much for you suggestion.

Neverthless, we think that including more details in the title would make it too long. However, if the editor deems it appropriate, we could consider to change it.

ABSTRACT and INTRODUCTION

Response:

The English of the Absctract and Introduction have been revised by the Mdpi’s English editor.

MM

Comment: “Experimental design. Since there are 4 groups (OO7, OO21, MF7, MF21), it is not necessary to list further definitions as this will confuse the reader about the exact number of groups, i.e. lines 90, 91: normal fat diet, NFD, OO7 and MF7, and 21% - high fat diets, HFD, OO21 90 and MF21. After these sentences, the reader is not sure about how many groups are ultimately involved in the experiment.

Response:

Thank you very much for your comment and proposal. The abbreviations NFD and HFD have been eliminated in this section.

Comment: “Section 2.4. the method should be written in more detail, because the method description is for other scientists to repeat the procedure given in the description”.

Response:

Thank very much for your suggestion. A brief description of the method has been included.

RESULTS

Comment: “For Table 2, I suggest that authors present the significantly changed variables in graphs rather than in a table because the visibility of differences is greater with a graphical presentation. Other non-significant results can be presented in the table”.

Comment: “The authors note that "there is a significant interaction between the two factors for all parameters (data not shown), suggesting that the effect of increasing fat content is different for each type of fat." Lines 225-226. I think it is important to define exactly for which type of fat the interaction between type of fat and percent fat is important. Please define it in the results along with the interaction effect in the graph/table for each dependent variable.

Response:

Thank very much for your interesting proposal. All variables, in tables 2 and 3,  that changed significantly with the hamsters’ diet are now presented in graphics. At the same time, significant differences between groups are expressed by superscript letters in order to show more clearly the interaction between the two factors, along with the explanation in the text.

Comment: “Table 4. as with table 2, I suggest that autohrs present significantly changed variables in graphs rather than tables because the visibility of differences is greater with a graphical representation. Other non-significant results can be presented in the table”.

Response:

Thank you very much for your valuable comment. However, in the case of Table 4, the number of variables (genes) is high and the significant differences found between groups are low. Therefore, in our opinion, the table shows quite clearly in which cases the effect of the factors is significant or not. We honestly believe that in this case the graph would not improve the visibility of the results.

DISCUSSION

The discussion is much too long. The authors should shorten it to a maximum of three pages. Please do not extensively describe the work of others to discuss your findings. One or two sentences with citations of other authors and one sentence with your opinion of that particular result are sufficient.

Response:

Thank you very much for your comment. We have revised the discussion many times and we can't find a way to significantly reduce the discussion by taking into account the criteria you suggest. We have removed some sentences that perhaps did not provide significant information, but even so, the discussion is longer than three pages. In any case, we have not found that the journal editorial imposes a limit on the length of the article, in general, and of the discussion in particular. Nevertheless, if the editors agree that we should reduce the discussion, we could do an extra effort, although we believe that this would mean a significant loss in the soundness of the discussion.

Comment: “Did the authors explain/discuss the result of the cluster analysis (lines 346-349)? How to explain the observed result when clustering the parameter AFST and plasma concentrations of TG, glucose, and DHA but not plasma cholesterol fractions?

Response:

In fact, we discuss the cluster analysis in a few occassions in the manuscript. For instance, in lines 1868-1870 , 1875-1877, 1970-1971 or 2162-2164 in the revised version. As we disscuss in lines 2162- “It has long been accepted that atherosclerosis is a multifactorial disease and that high-fat-diets, which cause hypertriglyceridemia and hyperglycemia, may contribute to its development [12]. Besides, we also mention that Martin et al. [25] also found that fasting plasma TC was not the best predictive factor for atherosclerosis in hyper-lipidemic hamsters. So, our results agree with what is mentioned by others, in the sense that it is a multifactorial disease that depends on various factors that are interrelated in a complex way, making it difficult to interpret and explain the results.

Reviewer 2 Report

The paper compares the effect of a diet with different content and type of fat (olive oil, sheep milk fat) on lipid metabolism indices. It was a good idea to use hamsters as a model to evaluate these changes. Numerous studies show that this model correlates well with atherogenic changes in humans, which the authors proved in their earlier work. The work is interesting, but as a reviewer, I have a few questions. When performing the statistical analysis, an analysis of variance was carried out, was it checked that the data met these conditions? What tests were used for this purpose? The authors wrote that they performed a logarithmic transformation of some data. It is worth highlighting which data it concerned.

The results and discussion are described appropriately. The conclusions are also correct.

Author Response

Thank you very much for reading the manuscript, for your work and for your  kind comments and suggestions to improve the work. We really appreciate it.

Comment:

“When performing the statistical analysis, an analysis of variance was carried out, was it checked that the data met these conditions? What tests were used for this purpose?

Response:

Thank you very much for your valuable question. Two new sentences have been added in 2.7 section answering to the questions.

Comment:

The authors wrote that they performed a logarithmic transformation of some data. It is worth highlighting which data it concerned.

Response:

Thank you very much for your suggestion. The following data were log-transformed: Some plasma fatty acids’ concentration (trans6-C16:1; vaccenic acid; α-linolenic acid and rumenic acid) and the expression level of HMG_CoAR gen. Nevertheless, we do not find necessary to include this information in the manuscript. We have consulted other works that carry out similar analyses and transformations and normally they do not use to give such detailed information, probably so as not to lengthen and complicate the explanation of the analysis too much.

Reviewer 3 Report

The study by Berriozabalgoitia et al. investigated the relationship between milk fat intake and the risk of atherosclerosis in Golden Syrian Hamster. The animals were fed two different fat diet (sheep milk fat or olive oil) for 2 weeks. The authors found that the type and percentage of fat affected plasma biochemical parameters related to lipid metabolism and the expression of five genes (CD36, SR-B1, ACAT, LDLR and HMG-CoAR).  Milk high fat based diet caused a more atherogenic plasma profile and both fats caused a similar increase in the thickness of the fatty streaks determined by aortic histology.

Comments

1.       The histological staining is of bad quality, the standard evaluation of the aortic pathology is missing.   The authors present atherosclerotic changes in aorta as the main readout of the study that is depicted in the title. Therefore, the evaluation of aortic pathology should be presented properly.

2.       Did you find any histological changes in the liver?

3.       The main significant serum parameters could be presented in Graphics.

4.       There are too many abbreviations in the text, please reconsider.

Author Response

Thank you very much for reading the manuscript, for your work and for making comments and suggestions to improve the work. We really appreciate it.

Comments:

The histological staining is of bad quality, the standard evaluation of the aortic pathology is missing.  The authors present atherosclerotic changes in aorta as the main readout of the study that is depicted in the title. Therefore, the evaluation of aortic pathology should be presented properly.

Response:

Thank you very much for your valuable comment. When we designed the study and the analyses to be carried out, we used as model some highly cited articles published in high-impact journals. For example:

Reference 46. Mangiapane, E.H.; McAteer, M.A.; Benson, G.M.; White, D.A.; Salter, A.M. Modulation of the regression of atherosclerosis in the hamster by dietary lipids: comparison of coconut oil and olive oil. Br J Nutr 1999, 82, 401-409. DOI: 10.1017/s0007114599001646 (29 citations) or

Reference 51: Mitchell, P.L.; Langille, M.A.; Currie, D.L.; McLeod, R.S. Effect of conjugated linoleic acid isomers on lipoproteins and atherosclerosis in the Syrian Golden hamster. Biochimica et Biophysica Acta (BBA) - Molecular and Cell Biology of Lipids 2005, 1734, 269-276, DOI 10.1016/j.bbalip.2005.04.007.(53 citations).

In these articles, as in our work, they histologically analyze fatty streak lesions as the only parameter to determine the extent of atherosclerosis.

Perhaps, the images shown in Figure 1 (Figure 4 in the new version) are not of the best resolution. This may be due to the fact that we have combined several photographs to obtain a single image and this could have led to a worsening of their quality. Actually, we obtained and analyzed many photographs. Following the protocol described by Mitchell et al., more than 80 sections of 10 µm we were prepared from each tissue sample and between 7 and 15 sections (those of best quality) from each animal were selected to do the measurements, so we could affirm that the obtained data have sufficient statistical strength.

As the study was carried out some years ago, it is impossible to carry out new analysis and to obytain new data, but, as explained before, this analysis has been long accepted for determining the risk of atherosclerosis in hamsters.

  1. Comment: Did you find any histological changes in the liver?

Response:

We did not perform histological analysis of the liver because we did not find large differences in hepatic lipid fractions.

  1. Comment: The main significant serum parameters could be presented in Graphics.

Response:

Done.

  1. Comment: There are too many abbreviations in the text, please reconsider.

Response:

Thank you very much for your suggestion. We have removed the less used abbreviations in the text.

Reviewer 4 Report

Firstly, I want to thank the authors for their interesting and well-written research.

While I was reading the article, I felt a shortage of two groups of hamsters, which were fed MF without VA and RA. As a result, the authors listed this in the study's limiting factors at the end of the article. In addition, it is good that the authors decided to put the hamsters in two animals in a cage, but it would be better if the animal sat alone. This will avoid the effect of collective eating.

I have a number of comments and questions. They are represented by points.

1.    Please indicate in paragraph 2.6. manufacturer for Oil Red O.

2.      Paragraph 2.7. Using parametric statistics in experimental biomedicine is illegal when there are 8 replicates per group. It is better and more correct to use non-parametric statistics, the Kruskal-Wallis test with the Dunn’s multiple comparison test. In this case, I recommend presenting the data in the form of a box and whiskers plot with the parameter «min to max. Show all points». The Boxplot has 5 characteristics: the minimum and the maximum (whisker borders), the sample median (located in the middle of the box), and the first and third quartiles (box borders). The replicates are represented by dots on the graph. The dots can be marked with color or different shapes. Boxplot visually characterizes the entire sample as a whole. Figures 1, 2, 3, and 5 show 8 measurements per group, if the reviewer understood correctly.

3.      Fig. 1, Fig. 3, Fig.5. I recommend authors do not use letters denoting significant differences. It is better to draw a line over the compared groups, above which the quantitative p value should be written. By doing this, you can immediately evaluate the presented data without referring to the figure caption or text for additional information.

4.      Fig.3. I recommend coloring triglycerides differently. The black color shows both total cholesterol and triglycerides. When you look, you focus on the fact that black is just total cholesterol. Also note comment 2.

5.       Fig. 5. In the caption to the figure, it is necessary to give explanations of the borders of the box (SD or min/max?), whiskers (SD or min/max?) and midline. Also note comment 2.

6.      Figures must be numbered consecutively. So, for example, after the number 5 comes the number 3, not 6. Please make the appropriate corrections to the text.

7.      Line 365. Liver enzymes may be listed in parentheses.

8.      Fig. 6. Please double-check the abbreviations of molecules and receptors in the figure and in the caption. If possible, give a transcript of all the names in the caption.

9.      I didn't see a sentence (or paragraph) mentioning groups OO7 and MF7 in the conclusion (or discussion). If the authors added this information, it would be great.

The article is easy to read and written clearly. Best wishes to the researchers for continued success.

Author Response

Thank you very much for reading the manuscript, for your work and for making comments and suggestions to improve the work. We really appreciate it.

Comment

While I was reading the article, I felt a shortage of two groups of hamsters, which were fed MF without VA and RA. As a result, the authors listed this in the study's limiting factors at the end of the article. In addition, it is good that the authors decided to put the hamsters in two animals in a cage, but it would be better if the animal sat alone. This will avoid the effect of collective eating.

Response:

Thank you very much for kind comments.

Regarding the number of animals per cage, we searched  in the scientific literature for similar studies with hamsters and found that in some cases hamsters were placed individually (e.i., references 6, 7, 8..), but also in colony cages (between 2 and 10 animals/cage (references 12,  24, 26, 47,..). In this study, when making the decision to place two hamsters per cage, we took into account that the cages in the animal facility were very large, that the hamsters were of the same sex, and that they were still young (4 weeks old). In addition, we placed more than one food and water bowls in each cage. We marked each animal to differenciate between them, and, at the end of the study we did not find big differences in the animal performances in each cage.

Nevertheless, we will take into account your suggestion for future studies.

Comment 2

Please indicate in paragraph 2.6. manufacturer for Oil Red O.

Response :

Done.

Comment:

Paragraph 2.7. Using parametric statistics in experimental biomedicine is illegal when there are 8 replicates per group. It is better and more correct to use non-parametric statistics; the Kruskal-Wallis test with the Dunn’s multiple comparison test. In this case, I recommend presenting the data in the form of a box and whiskers plot with the parameter «min to max. Show all points». The Boxplot has 5 characteristics: the minimum and the maximum (whisker borders), the sample median (located in the middle of the box), and the first and third quartiles (box borders). The replicates are represented by dots on the graph. The dots can be marked with color or different shapes. Boxplot visually characterizes the entire sample as a whole. Figures 1, 2, 3, and 5 show 8 measurements per group, if the reviewer understood correctly.

Response

Parametric data analysis has been accepted in animal studies, as we have verified in most of the works with hamsters that we have referenced. Nevertheless, we have applied the Kruskal-Wallis test, as suggested, and changed the P values accordingly in tables. As can be seen, practically the same significant differences previously found between groups are maintained.

Regarding the type of graphics, we highly appreciate your proposal. However, in the previous version of the manuscript we presented data in tables. Then, reviewers asked to change to graphs. We decided to use bars-graphs because, in our opinion, is the type of graph that more clearly shows the differences between groups, and is widely used among referenced studies (e.g., 10, 11, 12…). So, we don't see the need to change everything again.

Comment

Fig. 1, Fig. 3, Fig.5. I recommend authors do not use letters denoting significant differences. It is better to draw a line over the compared groups, above which the quantitative p value should be written. By doing this, you can immediately evaluate the presented data without referring to the figure caption or text for additional information.

Response

Thank you very much for your suggestion. We have tried to draw the lines over the compared groups, as you suggested, but realized that since we need to do the comparison between four groups, in pairs, we need to draw 6 lines over the bars, some of which need to cross, which makes it the interpretation very difficult. On the other hand, the use of different letters seems to us a very clear way of interpreting the results, even without having to read the figure caption.

Comment

Fig.3. I recommend coloring triglycerides differently. The black color shows both total cholesterol and triglycerides. When you look, you focus on the fact that black is just total cholesterol.

Response

Thank you very much for your valuable proposal. In fact, we have changed also the color of other parameters to avoid confusion.

Comment

Fig. 5. In the caption to the figure, it is necessary to give explanations of the borders of the box (SD or min/max?), whiskers (SD or min/max?) and midline. Also note comment 2.

Response

Thank you very much for your suggestion. The figure and the caption have been changed as suggested.

Figures must be numbered consecutively. So, for example, after the number 5 comes the number 3, not 6. Please make the appropriate corrections to the text.

Response

Thank you very much for noticing. We corrected it.

Comment

Line 365. Liver enzymes may be listed in parentheses.

Response

Liver genes which expression is upregulated in OO7 diet fed hamsters, in comparison with MF7 fed ones, are listed in parentheses.

Comment

Fig. 6. Please double-check the abbreviations of molecules and receptors in the figure and in the caption. If possible, give a transcript of all the names in the caption.

Response

Thank you for noticing. We have checked and added those that were missing. Only standardized abbrevitions for fatty acid were not added.

Comment

I didn't see a sentence (or paragraph) mentioning groups OO7 and MF7 in the conclusion (or discussion). If the authors added this information, it would be great.

Response

Thank you for your kind suggestion. In our opinion, the similarities and differences between these groups have  been explained and discussed in the results and discussion sections. So, we have introduced a new paragraph in the conclusions that explains the main differences found between the hamsters fed  OO7 and MF7 diet, and we have changed the section according to this new paragraph.

Comment

The article is easy to read and written clearly. Best wishes to the researchers for continued success.

Response

Thanks very much for your kind comments and wishes. We really appreciate them. 

Reviewer 5 Report

The article by Berriozabalgoitia et al describes the effect of Normal-fat vs. High-fat diets and Olive oil vs. CLA-rich Dairy 2 fat on atherosclerosis in Golden Syrian hamsters. The article is overall good written and interesting. However, some methodological concerns need to be solved/addressed before the manuscript can be published.

Major issues:

For olive oil and sheep milk – was the animal diet prepared from the same batches all the time, or different batches were used? If different batches were used how authors controlled for their consistency?

Hamsters are generally solitary animals, and can be aggressive towards other hamsters. Since hamsters were caged 2 at the time how it was ensured that both have equal access to food and water? What were the differences in food intake between 2 hamsters caged together? Was domination of one hamster over another observed?

Connected to previous, each cage, rather than each hamster, should be considered as data point for all statistical analysis (since it is impossible to judge if both hamsters in the same cage had equal access to food and water). It should also be considered for RNA levels measurement.

Points instead of bars should be presented on the graphs.

Figure 4 A-D. Original unmodified picture should be presented or added to supplementary.

Minor issues:

Figure 6 legend is named as Figure 3.

Author Response

The article by Berriozabalgoitia et al describes the effect of Normal-fat vs. High-fat diets and Olive oil vs. CLA-rich Dairy 2 fat on atherosclerosis in Golden Syrian hamsters. The article is overall good written and interesting. However, some methodological concerns need to be solved/addressed before the manuscript can be published.

Response:

Thank you very much for reading the manuscript, for your work and for making comments and suggestions to improve the work. We really appreciate it.

Comments

For olive oil and sheep milk – was the animal diet prepared from the same batches all the time, or different batches were used? If different batches were used how authors controlled for their consistency?

Response

As we explained in Materials and Methods, Harlan prepared the four basal mix diets taking into account the amount the fat  (including the cholesterol concentration)  to be added (milk fat or olive oil). Then we added the fat to the basal mixtures in order to get the four different diets. We prepared the mixtures at the beginning of the study and used the same mixture all along the study.

Comment

Hamsters are generally solitary animals, and can be aggressive towards other hamsters. Since hamsters were caged 2 at the time how it was ensured that both have equal access to food and water? What were the differences in food intake between 2 hamsters caged together? Was domination of one hamster over another observed?

Connected to previous, each cage, rather than each hamster, should be considered as data point for all statistical analysis (since it is impossible to judge if both hamsters in the same cage had equal access to food and water). It should also be considered for RNA levels measurement.

Response:

Thank you very much for kind comment. We will try to put only one hamster per cage in future studies. But fot this study, we searched  in the scientific literature for similar studies with hamsters and found that in some cases hamsters were placed individually (e.i., references 6, 7, 8..), but also in colony cages (between 2 and 10 animals/cage (references 12,  24, 26, 47,..). So,  when making the decision to place two hamsters per cage, we took into account that the cages in the animal facilty were very large, that the hamsters were of the same sex, and that they were still young (4 weeks old). In addition, we placed more than one food and water bowls in each cage. However, aware of the potential problems, the one-week adaptation period was in part intended to limit or minimise these drawbacks. We marked each animal to differenciate between them, and, at the end of the study we did not find big differences in the animal performances in each cage.

Cooment

Points instead of bars should be presented on the graphs.

Response

Regarding the type of graphics, we highly appreciate your proposal. However, in the previous version of the manuscript we presented data in tables. Then, reviewers asked to change to graphs. We decided to use bars-graphs because, in our opinion, is the type of graph that more clearly shows the differences between groups, and is widely used among referenced studies (e.g., 10, 11, 12…). So, we don't see the need to change everything again.

Comment

Figure 4 A-D. Original unmodified picture should be presented or added to supplementary.

Response

Thank you for you valuable suggestion. We have prepared a file with a selection of photographs, among the many that we have taken, to include them as supplementary material.

Minor issues:

Figure 6 legend is named as Figure 3.

Response

Done.

Round 2

Reviewer 1 Report

Generally, the way the paper is written is quite difficult to understand, and this is one of the main reasons why I think the paper is not of sufficient quality to be published in this journal. Apart from this general problem, I still think that the discussion is too long. And besides this, the main problem is that there is almost no discussion, only describing previous studies with references. Since this is not a review but an experimental paper, I must state that this kind of discussion part is not suitable for this kind of paper.

Although authors made some improvements, due to the overall average quality and soundness of the paper, my opinion is that paper still does not reach the priority high enough for publication in Metabolites journal. However, the ultimate decision lies upon the journal editors.

Author Response

Thank you very much for reading the manuscript and for giving your opinion. But obviously we can't agree with it.

In the previous round we made most of the changes the reviewer suggeted. The English of the manuscript has been reviewed by experts from the Mdpi editorial and the rest of the reviewers (4) comment that the manuscript is well written and easy to read.

Regarding the content and length of the discussion, as we commented on the previous round, we reduced it as long as we could without loosing its soundess, in our opinion. So, we agree with the reviewer that decision about this point lies upon the journal editors.

Reviewer 3 Report

Dear Editors, 

The authors improved the manuscript, it could be accepted for publication.

Best regards

Sincerely,

Elena Kaschina  

Author Response

Thank you very much for reading the manuscript, for your work and for your  kind comments.  We really appreciate it.